# Coral calcification in a changing World and the interactive dynamics of pH and DIC upregulation

Malcolm T. McCulloch[1,2], Juan Pablo D'Olivo[1,2], James Falter[1,2], Michael Holcomb[1,2] & Julie A. Trotter[1]

Coral calcification is dependent on the mutualistic partnership between endosymbiotic zooxanthellae and the coral host. Here, using newly developed geochemical proxies ($\delta^{11}B$ and B/Ca), we show that *Porites* corals from natural reef environments exhibit a close ($r^2 \sim 0.9$) antithetic relationship between dissolved inorganic carbon (DIC) and pH of the corals' calcifying fluid (cf). The highest $DIC_{cf}$ ($\sim \times 3.2$ seawater) is found during summer, consistent with thermal/light enhancement of metabolically (zooxanthellae) derived carbon, while the highest $pH_{cf}$ ($\sim 8.5$) occurs in winter during periods of low $DIC_{cf}$ ($\sim \times 2$ seawater). These opposing changes in $DIC_{cf}$ and $pH_{cf}$ are shown to maintain oversaturated but stable levels of carbonate saturation ($\Omega_{cf} \sim \times 5$ seawater), the key parameter controlling coral calcification. These findings are in marked contrast to artificial experiments and show that $pH_{cf}$ upregulation occurs largely independent of changes in seawater carbonate chemistry, and hence ocean acidification, but is highly vulnerable to thermally induced stress from global warming.

[1] Oceans Institute and School of Earth Sciences, The University of Western Australia, Crawley, Western Australia 6009, Australia. [2] ARC Centre of Excellence for Coral Reef Studies, The University of Western Australia, Crawley, Western Australia 6009, Australia. Correspondence and requests for materials should be addressed to M.T.M. (email: malcolm.mcculloch@uwa.edu.au).

Scleractinian corals together with their endosymbiotic dinoflagellates, *Symbiodinium* (zooxanthellae), have been spectacularly successful in building the tropical coral reef edifices that dominate many shallow-water environments and harbour more than one-third of the oceans' biodiversity. The ongoing viability of these iconic[1] tropical reef systems is however in question[2,3], with symbiont-bearing shallow-water corals now facing the combined challenge of both global warming and ocean acidification from rapidly rising levels of $CO_2$ (ref. 4). Critical to the success of reef-building corals is their ability to extract dissolved inorganic carbon (DIC) from seawater and efficiently convert it into calcium carbonate, the major constituent of their skeletons. While much progress has been made in identifying many of the key elements of the biologic machinery that are integral to the biocalcification process[5–7] (Fig. 1), there are still significant gaps in our understanding. Foremost is the relationship between declining seawater pH and its impact on pH upregulation of the coral's extracellular calcifying fluid[8–10], a process that occurs at least in part via Ca-ATPase pumping of $Ca^{2+}$ ions into the calcifying region in exchange for the removal of protons[11]. Of equal but largely overlooked importance, are the mechanisms via which the various pH-dependent species of DIC (that is, $CO_2$, $HCO_3^-$ or $CO_3^{2-}$) are produced, transported, and then inter-converted at the site of calcification. It has also long been recognized[12,13] that light plays a key role in driving rates of calcification, and that light-enhanced calcification occurs as a result of the photosynthetic activity of endosymbiont dinoflagellates (zooxanthellae), providing both energy and additional carbon needed to drive calcification. The exact mechanism(s) by which coral calcification is linked to endosymbiont photosynthesis has, however, remained largely enigmatic at the polyp scale (Fig. 1) the zooxanthellae are physically separated from the site of calcification[13–15] and, apart from pH, few direct measurements exist[16] of the chemical conditions necessary to constrain the biocalcification process. Here we provide new evidence for an intimate link between the biologically mediated process of $pH_{cf}$ upregulation of the calcifying fluid and biological control over the concentration of DIC in the calcifying fluid ($DIC_{cf}$). We find that over annual timescales there is an inverse correlation between $pH_{cf}$ and $DIC_{cf}$. This acts to maintain relatively stable levels of aragonite saturation in the calcifying fluid, and hence near-optimal rates of coral calcification, despite large seasonally driven variations in metabolically supplied DIC.

## Results

**Reef-water and coral calcifying fluid carbonate chemistry.** To reconstruct the carbonate chemistry of the calcifying fluid from which corals precipitate their aragonite skeleton, we use the boron isotopic composition ($\delta^{11}B$) as a proxy for the calcifying fluid $pH_{cf}$ (refs 10,17,18). For determining the carbonate ion concentrations $[CO_3^{2-}]_{cf}$ in the calcifying fluid, we use the combined $\delta^{11}B$-B/Ca proxy[19]. The application of the $\delta^{11}B$-B/Ca carbonate ion proxy has now been made possible by recent experimental measurements of the B/Ca carbonate ion distribution coefficient[19], a major limitation of previous studies[20] (see 'Methods' section). To examine how the chemistry of the calcifying fluid varies seasonally under 'real-world' reef conditions, we have analysed the skeletons of massive *Porites* collected from Davies Reef in the Great Barrier Reef and from Coral Bay Ningaloo Reef for which reef-water pH and sea-surface temperatures (SST) records are available[21,22] (see 'Methods' section). Species of massive *Porites* coral are ideal for reconstructing seasonal changes in the composition of their calcifying fluid since they are long-lived and, more importantly,

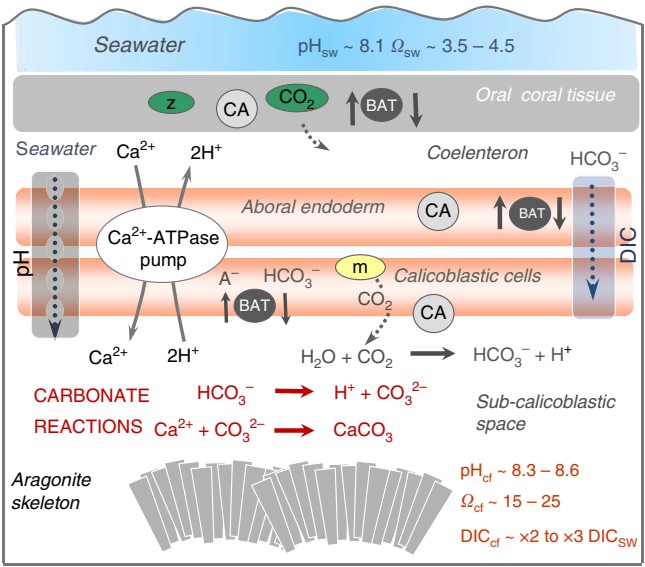

**Figure 1 | Mechanisms involved in coral calcification.** Calcification occurs within the subcalicoblastic space from an initial seawater-derived fluid with additional metabolic sourced supply of DIC[5–7]. Elevation of calcifying fluid $pH_{cf}$ occurs via removal of protons from the calcification site by $Ca^{2+}$-ATPase exchangers. The carbonic anhydrases (CA) catalyse the forward reactions converting $CO_2$ into $HCO_3^-$ ions[7,34]. Transfer of DIC into the subcalicoblastic space may occur via diffusion of $CO_2$ and/or by $HCO_3^-$ pumping via bicarbonate anion transporters (BAT)[5–7]. The link between the activity of zooxanthellae located in the oral coral endoderm tissue to the generation of metabolic DIC within the aboral endoderm and calicoblastic cells (orange) and transport to the calcifying fluid remains uncertain[5–7] (Figure modified from McCulloch et al.[31]).

the architecture of their skeleton has a relatively straightforward chronology that facilitates well-constrained timing of their skeletal growth at seasonal resolution[23]. Given that only limited records of seasonal changes in local seawater carbonate chemistry are available[22,24], these data are supplemented by model estimates[24] of the reef-induced pH variability. The Great Barrier Reef and Ningaloo Reef sites (see 'Methods' section) have a typical seasonal range in temperature from ~23 to 28 °C, as well as relatively narrow seasonal ranges in seawater $pH_{sw}$ (total scale) from ~8.02 in summer to ~8.08 in winter (Fig. 2). This limited seasonal range in average reef-water $pH_{sw}$ of ~0.06 pH units is comparable to that observed in the open ocean[25], a reflection of the tight balance between production and respiration[24] combined with the limited residence time of waters in most wave and tidally driven reef systems[21].

**Covariation of calcifying fluid $pH_{cf}$ and $DIC_{cf}$.** In contrast to the limited variation in reef-water $pH_{sw}$, we find that *Porites* colonies from both Davies Reef and Coral Bay exhibit strong seasonal changes in $pH_{cf}$, from ~8.3 during summer to ~8.5 during winter (Fig. 2). This represents an elevation in $pH_{cf}$ relative to ambient seawater of ~0.4 pH units together with a relatively large seasonal range in $pH_{cf}$ of ~0.2 units. These observations are in stark contrast to the far more muted changes based on laboratory-controlled experiments[9,17]. These inferred laboratory responses[10] in calcifying fluid pH ($pH^*_{cf}$) are shown in Figs 2 and 4, where the expected seasonal range is ~0.02 pH units, an order of magnitude smaller than those actually observed in reef environments. The explanation for this unexpectedly large range in seasonal $pH_{cf}$ present under natural reef conditions becomes apparent from the exceptionally strong and inverse

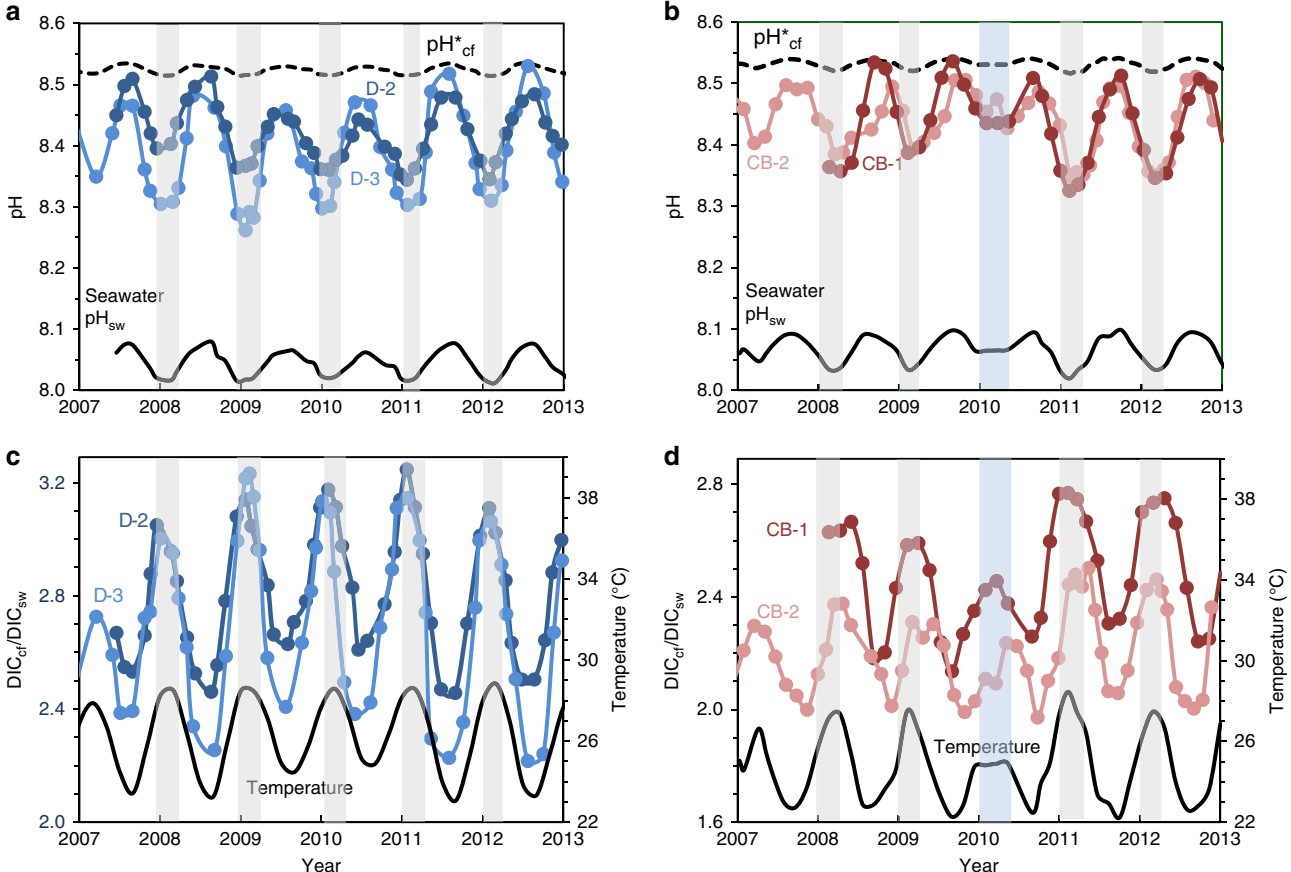

**Figure 2 | Seasonal time series of coral calcifying fluid pH$_{cf}$ and DIC$_{cf}$.** (**a**) *Porites* spp. coral calcifying fluid pH$_{cf}$ derived from $\delta^{11}$B systematics (see 'Methods' section and Supplementary Data) for colonies D-2 and D-3 from Davies Reef (18.8° S) in the Great Barrier Reef, Queensland. Shading denotes the summer period when pH$_{cf}$ and seawater pH$_{sw}$ values are at a minimum. Dashed line shows pH*$_{cf}$ expected from artificial experimental calibrations (pH*$_{cf}$ = 0.32 pH$_{sw}$ + 5.2)[10,17] with an order of magnitude lower seasonal range than measured pH$_{cf}$ values. (**b**) Same as previous for *Porites* colonies from Coral Bay (CB-1 and CB-2) in the Ningaloo Reef of Western Australia (23.2° S) showing seasonal fluctuations in pH$_{cf}$ and seawater pH$_{sw}$. The blue shading denotes the anomalously cool summer temperatures in 2010. (**c**) Enrichments in calcifying fluid DIC$_{cf}$ (left axis; coloured circles) derived from combined B/Ca and $\delta^{11}$B systematics together with synchronous seasonal variations in reef-water temperatures (right axis; black line) for *Porites* colonies from Davies Reef (GBR). The strong temperature/light control on DIC$_{cf}$ is consistent with enhanced metabolic activity of zooxanthellae symbionts in summer. (**d**) Same as previous but for *Porites* from Coral Bay (Ningaloo Reef, Western Australia).

correlations between pH$_{cf}$ and DIC$_{cf}$ ($r^2 = 0.88$–$0.94$) present at the colony level (Fig. 3a,b).

Here DIC$_{cf}$ reaches its highest values in summer ($\times 2.0$ to $\times 3.2$ higher than ambient seawater) and lowest values in winter, whereas pH$_{cf}$ shows the opposite pattern. This seasonal variability in DIC$_{cf}$ is consistent with light- and temperature-driven changes in the supply of metabolic DIC provided by the endosymbionts and/or the bicarbonate anion transporters within the coral host[7]. Thus, each *Porites* colony forms a distinctive subparallel array characterized by a distinctive range in DIC$_{cf}$ that is inversely correlated to pH$_{cf}$. Since the concentrations of carbonate ion [CO$_3^{2-}$] and consequently the aragonite saturation state ($\Omega_{cf} = $ [Ca$^{2+}$]$_{cf}$ [CO$_3^{2-}$]$_{cf}$/$K_{arag}$) of the calcifying fluid increases with increasing DIC$_{cf}$ and pH$_{cf}$, the observed antithetic seasonal changes in these parameters results in a more muted seasonal variation in $\Omega_{cf}$ ($\pm 5\%$ to $\pm 10\%$, Fig. 3a) compared to that expected from changes in only pH$_{cf}$ ($\pm 30\%$) or DIC$_{cf}$ ($\pm 12\%$) acting alone. While there remains a subdued positive correlation of $\Omega_{cf}$ with temperature (Fig. 3a), the inverse correlations between pH$_{cf}$ and DIC$_{cf}$ (Fig. 2a,b) indicate that the coral is actively maintaining both high ($\sim \times 4$ to $\times 6$ seawater) and relatively stable (within $\pm 10\%$ of mean) levels of elevated $\Omega_{cf}$ year-round.

While the absolute levels of enhanced $\Omega_{cf}$ are not dissimilar to previous qualitative estimates[17,26], the finding of significantly higher but relatively limited ranges in DIC$_{cf}$ of $\sim \times 2.0$ to $\sim \times 3.2$ seawater, is not generally consistent with recent micro-sensor[16] measurements. This difference may reflect the intrinsic limitations[6] of using probes that are 15–20 µm wide to measure the chemistry within the much narrower and irregular (1–10 µm) calcifying region. Additionally, separate probes are required for measurements of pH$_{cf}$ and [CO$_3^{2-}$]$_{cf}$, introducing further uncertainty, likely accounting for the large variability of *in situ* measured CO$_3^{2-}$ and hence inferred DIC$_{cf}$ ($\sim \times 1.4$ to $\times 4.2$ seawater). Finally and most importantly, regardless of the method employed, we find that measurements conducted under controlled, static, laboratory conditions[10] are unlikely to be representative of natural reef conditions due to the interactive dynamics of pH$_{cf}$ and DIC$_{cf}$ upregulation described herein.

**Discussion**

The underlying reason for the dynamic, antiphase relationship between pH$_{cf}$ versus DIC$_{cf}$, can be explained by the ability of the coral to 'control' what is arguably[27] one of its most fundamental

physiological processes, the growth of its skeleton within which it lives. For example, during winter (Fig. 2), there is a large systematic decrease in the abundance of metabolic DIC ($\sim 25\%$), presumably as a consequence of reductions in both light and temperature. Since higher pH shifts the carbonate equilibria to favour $CO_3^{2-}$ relative to $HCO_3^-$, the greater increase in $pH_{cf}$ in winter ($\sim 8.5$) compared to summer ($\sim 8.3$) increases the concentration of carbonate ions within the calcifying fluid (and therefore $\Omega_{cf}$) for the same $DIC_{cf}$. This increase in winter $pH_{cf}$ therefore partially counters the seasonal slowdown in host-symbiont carbon metabolism. Hence during the cooler periods, higher $pH_{cf}$ enhances $\Omega_{cf}$ and hence partially mitigates the reduced temperature-dependent kinetics of calcification because rates of mineral precipitation are proportional to $(\Omega-1)^n$, where $n$ is the temperature-dependent order of the reaction[28] ($n = 1.3$–$2.0$ for most reef habitats). During summer, the opposite behaviour is observed, with higher rates of metabolic $DIC_{cf}$ partially offset by decreases in $pH_{cf}$, resulting in a concomitant decrease in the carbonate saturation state of the calcifying fluid ($\Omega_{cf}$) and hence moderated (albeit still high) rates of calcification (Fig. 4c,d).

This implies that during summer, zooxanthellae-derived $DIC_{cf}$ is being supplied in excess of the 'optimal' requirements for the biologically mediated process of skeleton building. Thus, while existing mineral rate kinetics indicate that rates of calcification are still a factor of two- to fourfold higher in summer than in winter, this range is significantly less than the estimated eightfold higher summer rates (Fig. 4c,d) if constant levels of elevated $pH_{cf}$ upregulation were operative, as implied from the artificial constant seawater $pH_{sw}$ and temperature experiments[10].

Although our findings are based only on species of *Porites* from the Pacific and Indian Oceans, they nevertheless have important implications for our understanding of how reef-building corals in general will respond to climate change. The occurrence, for example, of the highest $pH_{cf}$ values during winter, when metabolically derived sources of energy are at a minimum, provides further evidence against the proposition that $pH_{cf}$ upregulation is an energetically costly process[29], and will therefore decline as seawater $pH_{sw}$ decreases due to ocean acidification. This is supported by results of the free ocean carbon enrichment experiment[30] conducted within the GBR Heron Island lagoon, where corals subjected to both natural and superimposed fluctuations in seawater $pH_{sw}$ exhibited essentially constant $pH_{cf}$ upregulation, a condition referred to by those authors[30] as 'pH homoeostasis'. These findings, combined with measurements of even higher $pH_{cf}$ in azooxanthellate deep-sea corals[31] ($pH_{cf} > 8.6$), are thus consistent with inferences that Ca-ATPase-driven $pH_{cf}$ upregulation is a relatively energetically inexpensive process[17]. These observations, in conjunction with the highly correlated and anti-cyclical seasonal changes in both $pH_{cf}$ and $DIC_{cf}$, therefore argue against the reduction of $pH_{cf}$ in summer being a result of the passive feedback from higher rates of calcification producing more protons thereby lowering $pH_{cf}$. Thus, while this possibility cannot yet be entirely excluded, the higher production rates of zooxanthellae-derived metabolites that are presumably available in the summer to facilitate enhanced Ca-ATPase activity, also suggest that the lower summer levels of $pH_{cf}$ is not due to intrinsic limitations in the Ca-ATPase $H^+$ pumping, but rather physiological controls on growth rate. Furthermore, similar anti-correlated changes in $pH_{cf}$ and $DIC_{cf}$ are present in *Porites* from both Davies and Ningaloo Reefs, despite large differences in growth rates.

Our findings also have major ramifications for the interpretation of the large number of experiments that have reported a strong sensitivity of coral calcification to increasing

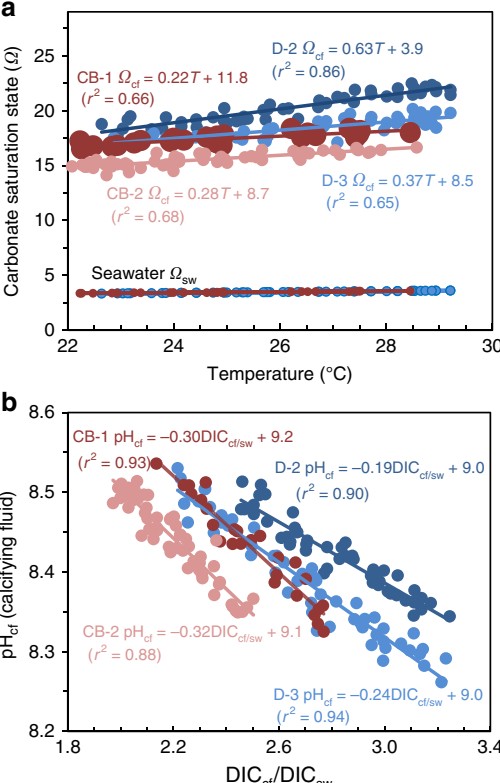

**Figure 3 | Covariation between calcifying fluid parameters $\Omega_{cf}$ versus seasonal temperature and $pH_{cf}$ versus $DIC_{cf}$.** (**a**) Covariation of calcifying fluid saturation state ($\Omega_{cf}$) with reef-water temperature showing a five- to sixfold elevation in $\Omega_{cf}$ relative to reef-waters for *Porites* corals from Davies Reef (D-2 and D-3) in the Great Barrier Reef and from Coral Bay (CB-1 and CB-2) in the Ningaloo Reef. Note the very narrow range ($\pm 5$ to $\pm 10\%$) of high $\Omega_{cf}$ values for each colony. (**b**) Subparallel arrays of inversely correlated ($r^2 = 0.88$-$0.94$) calcifying fluid $pH_{cf}$ and $DIC_{cf}/DIC_{sw}$ values reflecting specific bio-environmental controls at the colony level on metabolic $DIC_{cf}/DIC_{sw}$. Seasonal variations in metabolic supplied $DIC_{cf}$ are offset by opposing changes in $pH_{cf}$ that act to moderate the overall variations in $\Omega_{cf}$, the ultimate controller of skeletal growth rates.

ocean acidification[32]. An inherent limitation of many of these experiments[33] is that they were generally conducted under conditions of fixed seawater $pH_{sw}$ and/or temperature, light, nutrients, and little water motion, hence conditions that are not conducive to reproducing the natural interactive effects between $pH_{cf}$ and $DIC_{cf}$ that we have documented here. A characteristic common to a variety of coral species grown under these artificial conditions is the apparently constant but limited sensitivity (one-third to one-half) of $pH_{cf}$ relative to external changes in seawater $pH_{sw}$ (refs 10,17). While the reason for this apparently systematic but muted experimental response of $pH_{cf}$ is still uncertain, it likely involves reduced and/or constant levels of metabolically produced $DIC_{cf}$. Under such fixed conditions, we surmise that the supply of seawater DIC into the subcalicoblastic space (Fig. 1) becomes the dominant source and hence major influence on levels of $DIC_{cf}$, with upregulation of $pH_{cf}$ therefore acting as the major controller of $\Omega_{cf}$ and thereby affecting the perceived sensitivity of $pH_{cf}$ to ocean acidification. This inference is supported by the fact that the observed $pH_{cf}$ of *Porites* from both Davies and Ningaloo Reefs were closest to the $pH_{cf}$ predicted from the constant condition experiments in winter when $DIC_{cf}$ levels are naturally lowest due to reduced light and/or temperature, hence most similar to experimental predicted

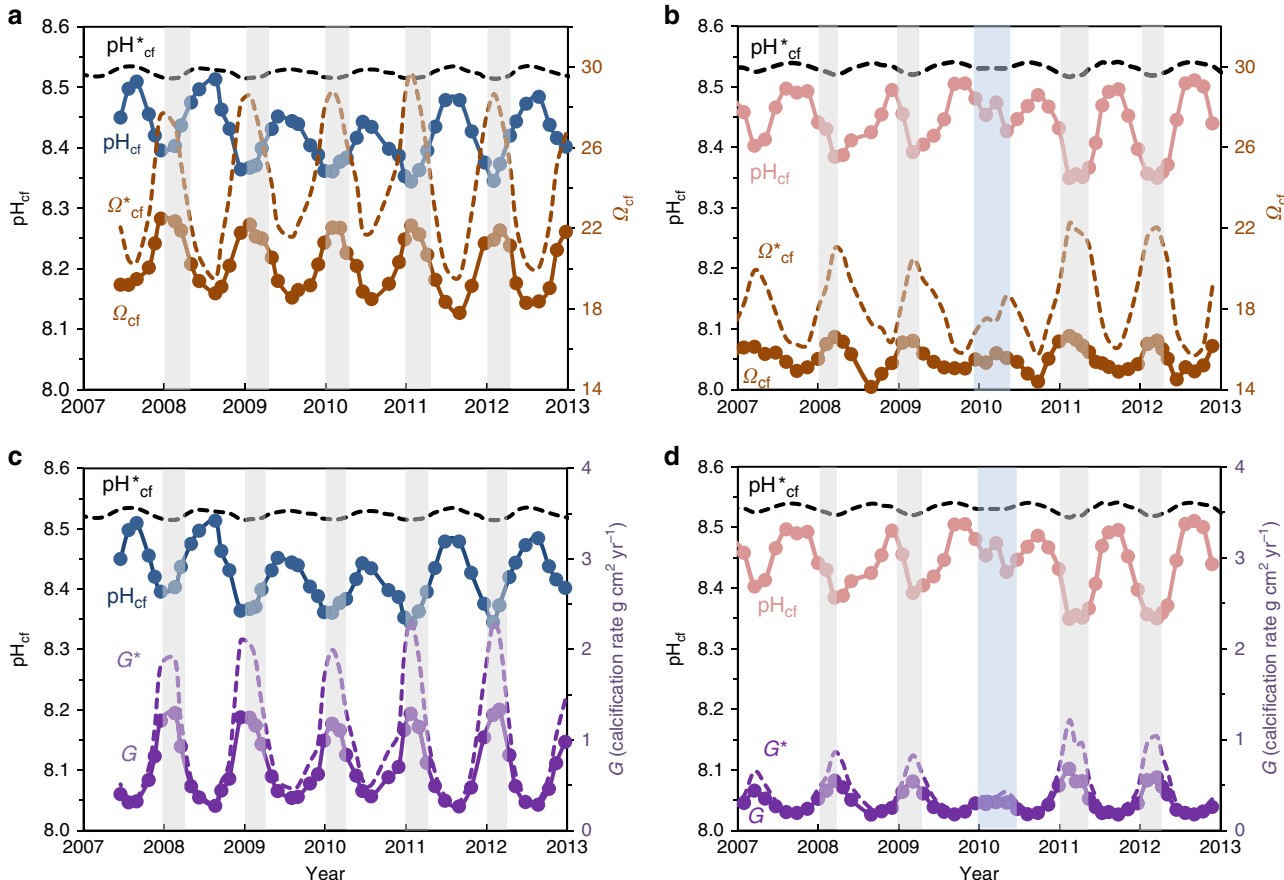

**Figure 4 | Seasonal time series of calcifying fluid pH$_{cf}$ and $\Omega_{cf}$ together with calculated calcification rates G.** (**a**) Calcifying fluid pH$_{cf}$ and $\Omega_{cf}$ values for *Porites* coral (D-2) from Davies Reef (GBR), where $\Omega_{cf} = [Ca^{2+}]_{cf}[CO_3^{2-}]_{cf}/K_{arag}$. Dashed line shows the $\Omega^{\star}_{cf}$ calculated using fixed experimental[10,17] pH$^{\star}_{cf}$ values (see Fig. 2a,b). (**b**) Same as previous for Coral Bay (Ningaloo Reef, Western Australia) *Porites* (CB-2). (**c**) Calcification rates calculated using the inorganic rate equation[28] $G = k(\Omega - 1)^n$, where $k$ and $n$ are the temperature-dependent constant and order of the reaction, respectively[28]. Because of opposing changes in pH$_{cf}$ relative to DIC$_{cf}$ (Fig. 1), $\Omega_{cf}$ and hence coral growth rates are strongly modulated reducing seasonal variations by twofold compared to those estimated from fixed condition experiments ($G^{\star}$). (**d**) Same as previous for *Porites* from Coral Bay (Ningaloo Reef, Western Australia).

seawater end-member values. Clearly, since the interactive dynamics of pH$_{cf}$ and DIC$_{cf}$ upregulation do not appear to be properly simulated under the short-term conditions generally imposed by such artificial experiments, the relevance of their commonly reported finding of reduced coral calcification with reduced seawater pH must now be questioned.

In summary, we have now identified the key functional characteristics of chemically controlled calcification in reef-building coral. The seasonally varying supply of summer-enhanced metabolic DIC$_{cf}$ is accompanied by dynamic out-of-phase upregulation of coral pH$_{cf}$. These parameters acting together maintain elevated but near-constant levels of carbonate saturation state ($\Omega_{cf}$) of the coral's calcifying fluid, the key driver of calcification. Although the maintenance of elevated but near-constant $\Omega_{cf}$ in mature coral colonies is not directly influenced by ocean acidification, it is however highly susceptible to thermal stress. In extreme cases of coral bleaching, the loss of endosymbionts disrupts the metabolic supply of DIC$_{cf}$ as well as the metabolites necessary to operate the Ca-ATPase that upregulate pH$_{cf}$ (refs 14,34), thus effectively terminating calcification. So, although rising levels of $p_{CO_2}$ can have adverse effects on the recruitment and growth of juvenile corals[35–38], especially those lacking robust internal carbonate chemistry regulatory mechanisms, extreme thermal stress is detrimental to all symbiont-bearing corals[39,40] regardless of their growth stage.

We therefore conclude that the increasing frequency and intensity of coral bleaching events due to CO$_2$-driven global warming constitutes the greatest immediate threat to the growth of shallow-water reef-building corals, rather than the closely associated process of ocean acidification.

## Methods

**Reef sites.** *Porites* colonies were sampled from two reef systems: (1) Davies Reef (18.8° S, 147.63° E), a mid-shelf reef ∼100 km east-northeast of Townsville, Queensland, Australia in the central Great Barrier Reef, and (2) Coral Bay (23.19° S, 113.77° E), part of the Ningaloo Reef coastal fringing system of Western Australia. At Davies Reef, the annual range of daily average SST is 23–28.5 °C with a diurnal range of ∼0.5 °C or less[41]. *In situ* seawater temperature data extending back to 1987 for the core site at Davies Reef (18.83° S, 147.63° E) was compiled from a number of different temperature sensors deployed between a depth of ∼2 to ∼10 m maintained by the Australian Institute of Marine Science from October 1991 to December 2013 (http://data.aims.gov.au/aimsrtds/datatool.xhtml). To estimate seasonal changes in carbonate chemistry, we used the 24-h seawater carbonate chemistry data collected by Albright et al.[22] on the lagoon side of the Davies Reef flat around the summer and winter extremes in both light and temperature. Their data showed that the daily average pH at that reef site was 8.02 in summer and 8.08 in winter; a seasonal range that was similar to seasonal minima and maxima observed and hind-cast at Coral Bay and hence similar to what would be expected from seasonal variations in temperature-driven $p_{CO_2}$ solubility. We therefore assumed that daily average pH at Davies Reef also followed seasonal changes in temperature according to pH$_{sw}$ = − 0.010 × T + 8.31.

At Coral Bay, SST generally ranges from 22–23 °C in winter to 27–28 °C in summer[21]. To hind-cast seasonal changes in reef-water temperature and pH, we first used time series of SST data from just offshore Coral Bay at ∼25 km

resolution produced by Reynolds et al.[42] before June 2010 and then at ~1 km resolution produced by Chao et al.[43] Both SST data products were then calibrated against in situ observations of temperature collected from a moored depth of ~17 m as described by Falter et al.[21] and previous model studies of wave-driven circulation. The carbonate chemistry of Coral Bay and offshore waters (~2 km) were monitored between May 2011 and June 2012 and intermittently since then, with seasonal changes in offshore seawater $pH_T$ (total scale) being found to be strongly correlated with seasonal changes in offshore temperature ($pH_{sw} = -0.012 \times T + 8.37$, $r^2 = 0.86$, $n = 13$). To determine seasonal changes in pH at the back-reef site where the coral cores were recovered, the offshore pH was adjusted to account for the deviation in temperature due to local heating and cooling (see above), as well as the daily average decrease in total alkalinity of ~10 μmol kg$^{-1}$ at back-reef sites observed from measurements[44].

**Boron isotopic pH proxy.** Changes in the isotopic ratio of $^{11}B$ (~80%) and $^{10}B$ (~20%) are expressed in delta notation (in per mil, ‰) as:

$$\delta^{11}B_{carb} = \left[ \left( ^{11}B/^{10}B_{carb} / ^{11}B/^{10}B_{NIST951} \right) - 1 \right] \times 1{,}000, \qquad (1)$$

where $^{11}B/^{10}B_{carb}$ is the isotopic ratio measured in the coral carbonate and $^{11}B/^{10}B_{NIST951}$ is the isotopic ratio of the NIST SRM 951 boric acid standard. In seawater, boron exists as two different species, boric acid (B(OH)$_3$) and the borate ion (B(OH)$_4^-$), with their relative abundance being pH dependent. The sensitivity of the $\delta^{11}B$ proxy to the calcifying fluid $pH_{cf}$ arises from the incorporation of only the borate ion species into the aragonite structure[45–47], with the $\delta^{11}B$ isotopic composition reflecting the pH sensitivity of the borate versus boric acid speciation. The pH of the calcifying fluid ($pH_{cf}$) can thus be calculated from the $\delta^{11}B$ measured in the coral carbonate ($\delta^{11}B_{carb}$). The equation used to convert the $\delta^{11}B_{carb}$ isotopic composition measured in the coral carbonate skeleton to a pH of the calcifying fluid ($pH_{cf}$) is given by[48]:

$$pH_{cf} = pK_B - \log\left[ \frac{\left( \delta^{11}B_{sw} - \delta^{11}B_{carb} \right)}{\left( \alpha_{(B3-B4)}\delta^{11}B_{carb} - \delta^{11}B_{sw} + 1{,}000\left( \alpha_{(B3-B4)} - 1 \right) \right)} \right], \qquad (2)$$

where $\delta^{11}B_{sw}$ represents the $\delta^{11}B$ in seawater ($\delta^{11}B_{sw} = 39.61‰$)[49] and $\alpha_{(B3-B4)} = 1.0272$ (ref. 50). The dissociation constant of boric acid $pK_B$ has a well-established value of 8.597 at 25 °C and a salinity of 35 (ref. 51). Here we also assume that the calcifying fluid has the same $\delta^{11}B$ composition as seawater since that is the ultimate source of boron and, due to the low $K_D$ of B/Ca (ref. 19), the boron composition and concentration of the calcifying fluid remains essentially constant during calcification. Recent studies utilizing the $\delta^{11}B$ $pH_{cf}$ proxy as well as direct measurements of calcifying fluid pH using pH-sensitive dyes[9,18], have also confirmed that under highly controlled artificial conditions of constant pH and temperature, corals upregulate the $pH_{cf}$ of their calcifying fluid by $^1/_3$ to $^1/_2$ relative to ambient seawater pH.

**B/Ca constraints on calcifying fluid DIC concentrations.** Prior studies indicate that borate rather than boric acid is the predominant species occupying the lattice position normally taken up by the carbonate ion[52] in calcifiers that precipitate aragonite skeletons. Although there are a number of reaction pathways through which this substitution could occur[19,20], it is likely to involve de-protonation of the borate species to create a divalent base ion with the same charge as that of the carbonate ion species (−2), to preserve the charge neutrality of the growing crystal:

$$Ca^{2+} + B(OH)_4^- = CaH_3BO_4 + H^+. \qquad (3)$$

The partitioning of borate versus carbonate into aragonite is thus likely to be sensitive to solution pH[10,19,20]. Here the relevant partition coefficient $K_D$ relating the molar ratio of (B/Ca)$_{CaCO_3}$ to the concentrations of the carbonate $[CO_3^{2-}]_{sol}$ and borate $[B(OH)_4^-]_{sol}$ species in the precipitating solution is determined using:

$$K_D \equiv (B/Ca)_{CaCO_3} \times \frac{[CO_3^{2-}]_{sol}}{[B(OH)_4^-]_{sol}}. \qquad (4)$$

Holcomb et al.[19] conducted experiments quantifying the ratio of boron to calcium in aragonite precipitated inorganically under a wide range of carbonate chemistries (including pH) and total DIC and boron concentrations, as well as conditions of pH and DIC appropriate to those in the calcifying fluid of corals. Furthermore, Holcomb et al.[19] also showed the close relationships between B/Ca, $CO_3^{2-}$ and $K_D$ based on substitution reactions between B(OH)$_4^-$ and $CO_3^{2-}$. Re-analysing the Holcomb et al.[19] data, we find (Fig. 5) that the observed $K_D$ as defined in equation (4) shows the expected decrease as a function of the concentration of total active protons within the precipitating solution.

Thus, using the definition of $K_D$ from equation (4) and its dependency on $pH_{cf}$ as given by the inorganic data of Holcomb et al.[19], we can now calculate the concentration of carbonate ions within the calcifying fluid (that is, $[CO_3^{2-}]_{cf}$ from measurements of (B/Ca)$_{carb}$ and $pH_{cf}$, the latter derived from the skeletal boron isotopic ratio ($\delta^{11}B_{carb}$). We further assume that $[B_T]_{cf}$ is equal to the total concentration of boron of ambient seawater and only a function of seawater salinity

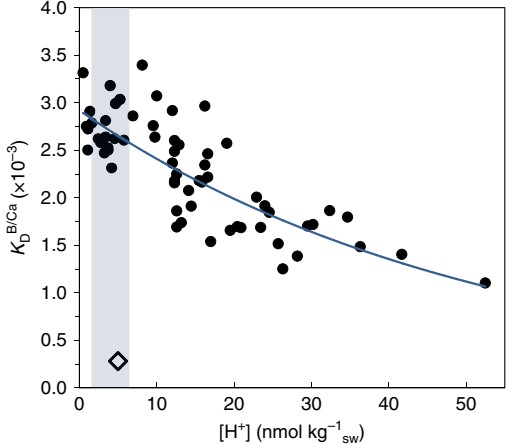

**Figure 5 | Experimentally determined B/Ca partition coefficient as a function of hydrogen ion concentration.** Measured B/Ca partition coefficient ($K_D$) as defined by equation (4) from the data of Holcomb et al.[19] The line represents a best-fit exponential curve to the data with $K_D^{B/Ca} = K_{D,0} \exp(-k_{K_D}[H^+]_T)$, where $K_{D,0} = 2.97 \pm 0.17 \times 10^{-3}$ (±95% CI), $-k_{K_D} = 0.0202 \pm 0.042$, $r^2 = 0.64$ and $n = 63$. The range for $pH_{cf}$ of upregulating calcifiers (that is, Porites spp.) is between ~8 and ~9 (shaded); equivalent to $[H^+]_T$ of between 1 and 10 nmol kg$^{-1}$ giving a range in $K_D^{B/Ca}$ ($\times 10^{-3}$) of 2.6–2.8, and therefore relatively in-sensitive to changes in coral $pH_{cf}$. Importantly, our experimentally determined $K_D^{B/Ca}$ value is an order of magnitude higher than the previous estimate by Allison et al.,[20] (open diamond) and consistent with the substitution of B(OH)$_4^-$ with $CO_3^{2-}$.

($[B_T]_{cf} = [B_T]_{sw}$ at salinity = 35). We therefore have:

$$\left[ CO_3^{2-} \right]_{cf} = K_D \times \left[ B(OH)_4^- \right]_{cf} \Big/ (B/Ca)_{CaCO_3} \qquad (5)$$

Where $K_D = 0.00297\exp(-0.0202[H^+]_T$ and for typical calcifying fluid $pH_{cf}$ values $K_D \sim 0.0027$, an order of magnitude higher than a previous estimate[20]. The concentration of DIC within the calcifying fluid is then calculated from the measured $pH_{cf}$ (equation 1) and $[CO_3^{2-}]_{cf}$ (equation 2) values using the programme CO2SYS provided by Lewis and Wallace[53], with the carbonate species dissociation constants of Mehrbach et al.[54] as re-fitted by Dickson and Millero[55], the borate and sulfate dissociation constants of Dickson[51,56], and the aragonite solubility constants of Mucci[57]. We also note that our use of a reliable experimentally determined $K_D$ is now consistent with substitution of borate with carbonate ion, rather than the previously inferred[20] substitution with bicarbonate ion, the latter assumption effectively negating the role of carbonate saturation state on calcification.

**Data availability.** The coral geochemical and seawater carbonate chemistry and temperature data are available in Supplementary Data.

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

## Acknowledgements

This research was supported by funding provided from an ARC Laureate Fellowship (LF120100049) awarded to Professor Malcolm McCulloch and the ARC Centre of Excellence for Coral Reef Studies (CE140100020). Measurements of the $\delta^{11}$B isotopic and B/Ca elemental ratios were conducted at the University of Western Australia's Advanced Geochemical Facility for Indian Ocean Research (AGFIOR), and we thank Anne-Marin Comeau and Dr Kai Rankenburg for their technical assistance.

## Author contributions

M.T.M. wrote the draft of the manuscript and all authors (M.T.M., J.P.D., J.F., M.H. and J.A.T.) participated in collecting the geochemical data, analysing the results and shaping the final manuscript.

**Additional information**

**Competing interests:** The authors declare no competing financial interests.

**Publisher's note**: 

