## [Peer Review File · Nature Communications]

Reviewers' Comments:

Reviewer #1 (Remarks to the Author)

Coral reefs are facing multiple stresses of global warming and ocean acidification (OA). Many works have addressed how corals may respond to OA with a focus on a mechanistic understanding of coral calcification. It is still quite debatable what is the interior pH and DIC in the calcifying fluid (cf) due to the challenge of the work and limitations of various approaches. This work represents an important progress and is novel and timely.

1. This paper uses $\delta^{11}\text{B}$ to derive pH_{cf} and B/Ca geochemical tracer to derive $[\text{CO}_3^{2-}]$ in corals under natural ocean conditions (not cultured/controlled). These authors find that pH_{cf} is about 8.3 to 8.5 (similar to the Allison et al. 2014 paper), which is much lower than the recent report by Cai et al. (2016) based on microelectrodes. Their reported $[\text{CO}_3^{2-}]$ (I guess from omega) are actually quite similar to the microelectrode results. The rest is CO₂SYN calculation. With a high pH_{cf} the same $[\text{CO}_3^{2-}]$ (and omega too) predicts low DIC_{cf} while the opposite is true for low pH_{cf}. Thus these authors find their DIC_{cf} to be 2x-3x of the seawater values. This is not totally new as the Allison et al (2014) paper says similarly. What is novel of this work is that data were obtained from outdoor in the natural reefs and in that anti-phase changes in pH_{cf} and DIC_{cf} were observed. I feel the authors need to note in the paper clearly the differences and similarities with previous work. In doing so I believe it is necessary to note that the boron isotope work represents an average of light-dark conditions and even some level of seasonal average while the microelectrode represents a short time scale of light period. Also the microelectrode only measures the cf space under the coral mouth while the geochemical tracers measure average of corals. The authors used the work of Venn et al. (2011) to support their lower pH_{cf}. But they should also recognize the differences between microelectrode and pH-dye based measurements. For example, the former was made through the center of coral polyps that were situated at the apex of larger colonies, while Venn et al. worked on the edge of laterally, newly growing microcolonies of coral on a glass slide. There can be significant differences in coral physiology between the top and sides/edges of most corals, including differences in calcification rates, symbiont density, and isotopic values

2. I feel one critical question is whether the $[\text{CO}_3^{2-}]$ cf geochemical tracer method is totally independent from the pH_{cf} tracer method. In p.2 (Results), the paper says "The $\delta^{11}\text{B}$ acts as a proxy for pH_{cf} while the newly developed combination of B/Ca and $\delta^{11}\text{B}$ now provides a proxy for the carbonate ion concentration $[\text{CO}_3^{2-}]$ cf of the calcifying fluid" According to this, it seems to me that these approaches are not independent; they both rely on $\delta^{11}\text{B}$ data. If so, then, the assumptions in deriving pH_{cf} are carried over into the $[\text{CO}_3^{2-}]$ cf. I checked the equations in p.10 and the Chemical Geology paper and do not see that pH_{cf} is needed in calculating $[\text{CO}_3^{2-}]$. Is this because both KD and $[\text{B}(\text{OH})_4^-]$ cf contains/requires pH_{cf}? Then, please comment if pH_{cf} is underestimated, which direction will it affect the estimation of $[\text{CO}_3^{2-}]$ cf and DIC_{cf}? Some explanation/clarification is needed here.

I remember in the Allison et al. 2014 paper, when they used the B/Ca tracer to derive DIC, they had to make an assumption whether B is precipitated into the CaCO₃ with HCO₃⁻ or with CO₃²⁻. When the assumption is the precipitation with HCO₃⁻, one would get high DIC, but if it is with CO₃²⁻, one would get low DIC. I suspect this is a related issue if indeed the $[\text{CO}_3^{2-}]$ is derived as a combination of B/Ca and $\delta^{11}\text{B}$.

3. At the relatively low pH_{cf} and the 2-3X DIC, what is the $[\text{CO}_2]$ concentration inside the cf? Is it still realistic that enough respiratory CO₂ can diffuse from the coelenteron into the cf? The very high internal DIC (low pH_{cf}) indicates relatively high $[\text{CO}_2]$ inside the cf. Thus, does CO₂ molecular still have the needed gradient to diffuse in to supply the calcification rate? The authors should check if the high DIC result is internally consistent.

Reading notes (or minor points)

p.1, while nothing really wrong, it's first time I heard anyone say "dissolved inorganic carbonate

(DIC)". We normally say dissolved inorganic carbon (DIC) though the meaning is the same that is the total concentration of all dissolved inorganic carbon species. I don't think it is necessary the authors create new name.

p.1 and several other places: the citation # after CO₂ should be CO₂ ref14 (not CO₂14) to avoid confusion.

p.1, for Ca ions, a "2+" should be attached to it (Ca²⁺)

p.1. fix this "It has also been long been recognized"

figure 1 caption, delete repeat: Transfer of DIC into the into the

Figure 1. although the fig caption says "Transfer of DIC into the into the subcalicoblastic space may occur via diffusion of CO₂ and/or by HCO₃⁻ pumping via bicarbonate anion transporters" the graphic illustration (right middle part) only shows HCO₃⁻ transport. The CO₂ diffusive transport is not clear. This should be the main transport DIC into the coral interior space to support calcification as argued by Allison et al. 2014 and Cai et al. 2016.

p.2, Results

the paper says "The δ¹¹B acts as a proxy for pH_{cf} 5,7 while the newly developed⁷ combination of B/Ca and δ¹¹B now provides a proxy for the carbonate ion concentration [CO₃²⁻]_{cf} of the calcifying fluid"

My question is—do you do both B/Ca and δ¹¹B to derive [CO₃²⁻]? Or just B/Ca? If you need both, then, I would argue the two methods are not strictly independent, that is all the assumptions in deriving pH_{cf} are carried over into the [CO₃²⁻].

I checked the equation in p.10 (it should also be numbered) and the Chemical Geology paper and do not see pH_{cf} as a needed information in calculating [CO₃²⁻].

p.4, top lines: "The explanation for this unexpectedly large range in seasonal pH_{cf} found under 'real-world' conditions becomes especially apparent" this sentence needs modification as I do not see any "explanation" let alone it be "apparent". If what follows "The DIC_{cf} thus also shows" provides the explanation, then, some transition words should be used and "thus" should be deleted.

p.4. you said "Whilst the absolute levels of enhanced Ω_{cf} are not dissimilar to previous estimates^{4,28}, the finding of significantly higher (~x2 to x3 seawater) DIC_{cf} is not consistent with recently reported micro-sensor CO₃²⁻ measurements²² pointing to limitations of such intrusive measurements into the tight sub-micrometric (1-10 μm) calcifying region²." I think some clarification is needed here. First, as far as [CO₃²⁻] (or Ω_{cf}) is concerned, the range reported here (4 to 6 times of seawater) is not different from the direct microelectrode measured [CO₃²⁻] reported in ref 22 by Cai et al. 2016. What is really different is pH_{cf}. This fact should be pointed out. Because of a much higher pH_{cf} in Cai et al., their calculated DIC_{cf} is very low (1X seawater). Second, what Cai et al. reported in lab was under light condition and nearly instantaneous. This can be very different from the seasonal averaged pH_{cf} reported here. I agree there are limitations of the microelectrode approach, but simply say "pointing to limitations of such intrusive measurements into the tight sub-micrometric (1-10 μm) calcifying region²" is probably not enough. Possible real differences should be mentioned.

p.5, bottom, "Together these observations also argue against the possibility that the reduction of pH_{cf} in summer is due to the passive feed-back from higher rates of calcification producing more protons thereby lowering pH_{cf}. Whilst this possibility cannot yet be entirely excluded the higher production rates of zooxanthellae derived metabolites in the summer to drive Ca-ATPase activity also suggest that the season up-regulation of elevated pH_{cf} is due direct to growth rate control rather than from limitations in the Ca-ATPase H⁺ pumping."

I just don't see how you can distinguish one from the other. But you can use the ratio of changes in DIC and TALK to say something.

p.x, 6 lines above Methods: "Although this process is not directly influenced by ocean acidification, it is however highly vulnerable to thermal stress." To say a process is "vulnerable" doesn't really

sound right. modify.

Refs:

Allison, N.; Cohen, I.; Finch, A. A.; Erez, J.; Tudhope, A. W., Corals concentrate dissolved inorganic carbon to facilitate calcification. *Nat Commun* 2014, 5.
Cai, W.-J.; Ma, Y.; Hopkinson, B. M.; Grotto, A. G.; Warner, M. E.; Ding, Q.; Hu, X.; Yuan, X.; Schoepf, V.; Xu, H.; Han, C.; Melman, T. F.; Hoadley, K. D.; Pettay, D. T.; Matsui, Y.; Baumann, J. H.; Levas, S.; Ying, Y.; Wang, Y., Microelectrode characterization of coral daytime interior pH and carbonate chemistry. *Nat Commun* 2016, 7.
Venn, A.; Tambutté, E.; Holcomb, M.; Allemand, D.; Tambutté, S., Live tissue imaging shows reef corals elevate pH under their calcifying tissue relative to seawater. *PLoS ONE* 2011, 6, (5), e20013.

Reviewer #2 (Remarks to the Author)

This article focuses on the active control of the coral host and photo symbionts on the chemical properties (DIC and pH) of the calcifying fluid (cf) under natural conditions. Using state of the art methods: Boron isotopic pH proxy and B/Ca constraints on calcifying fluid DIC concentrations, the authors reconstructed the latter properties in correlation to the natural conditions at sea.

Main find: antithetic relationships between DIC (dissolved inorganic carbon) and pH of the corals calcifying fluid (cf). while in the summer, under higher temperatures DIC_{cf} was 3.2 seawater the pH was 8.3, in the winter DIC_{cf} was 2.0 seawater the pH was 8.5. In both cases maintaining $\Omega_{cf} \sim 5$ seawater.

The conceptual main conclusion and in my eyes the most important is: "These findings are in marked contrast to artificial experiments and show that up-regulation of pH_{cf} occurs largely independent of changes in seawater carbonate chemistry and hence ocean acidification."

General comments to authors: this work is of the utmost high end both technically and conceptually to try and elucidate the different biochemical and chemical pathways that influence the calcification process in corals. It is very well presented and the writing allows a wide range of readers to take part in this fascinating discussion. It was indeed a pleasure to read and I am happy if my comments will augment this article in its final form.

Comments to the authors

First paragraph:

Not sure if the authors meant to put in the word "but" in the sentence perhaps "and" is also ok: "These opposing changes in DIC_{cf} and pH_{cf} are shown to maintain highly oversaturated ($\Omega_{cf} \sim 5$ seawater) but relatively stable levels of carbonate saturation, the key parameter controlling the rate of coral calcification"

In the following sentence where the authors make the claim "These findings are in marked contrast to artificial experiments a..." I think that the authors should list the references to these articles.

Second section / intro and the rest

My main issue with this section is rooted in the described (fig 1) mechanism for DIC transport and pH homeostasis. The authors chose to calculate the pH in the calcifying fluid (pH_{cf}) using the $\delta^{11}B$ value of sea water - $\delta^{11}B$ value of the coral skeleton. This calculation excludes the effect of the diffusion boundary layer (DBL) on the isotopic composition of the actual pool the coral incorporates into its skeleton. In 1995 Michael Kuhl et al. published a paper (*Mar. Ecol. Prog. Ser.*, Vol. 117: 159-172, 1995.) where for the first time he introduced a pH profile of the DBL. This publication followed an earlier one by Shashar et al. 1993 *Biol. Bull.* 185: 455-461 (1993), who presented the oxygen gradient in a coral DBL. In 2005 El Horani published another work that followed the DBL under changing temperatures establishing how in close proximity to the corals surface the changes in pH are pronounced (a Δ of 0.9 pH units). Lastly a paper from the Tchernov (*Biogeosciences*, 12, 5677-5687, 2015) also showed the effect of symbiont photosynthesis rates

on the DBL pH and finally on the $\delta^{11}\text{B}$ value of the coral skeleton.

The pH near the coral surface will influence the pool of Boron that will finally be incorporated via the cf into the skeleton. The pH in the cf will further influence the Boron isotopic composition via discrimination based on the local pH. Therefore we are facing a two-step discrimination process (as I understand).

In the methods section equation 2 is presented: (cant paste it here) but I am sure you have it :)

$\delta^{11}\text{B}_{\text{sw}} = 39.61 \text{ ‰}$ is referred from (Foster, G. L., Pogge von Strandmann, P. A. E. & Rae, J. W. B.) as a fixed value thus not considering the boundary layer (depending on the non-constant photosynthesis \ respiration rates as described by relevant literature). From this equation alone it is evident that marked changes in seawater Boron isotopes values will affect the pH_{cf}.

With all that said, I still think that the work presented merits publication in this prestigious journal as it shows the mechanistically coupled alga -host biochemical and chemical processes influencing the calcification pathway in corals. The summer and winter differences are quite convincing and are indeed a valuable contribution to our field.

From a plethora of manuscripts we gather that during summer the photosynthetic rate is enhanced in agreement with the authors statements ("was found during summer growth, consistent with thermal/light enhancement of metabolically (zooxanthellae) derived DIC, while the highest pH_{cf} (~8.5) occurred in winter during periods of low DIC_{cf} (~2 seawater).") that does imply that the coral does indeed actively controls the cf's pH otherwise it would be in direct correlation with temperature (high temperature high Photosynthetic rate leading to high cf pH) however the authors show a reverse situation. We still have to consider that this was calculating with a fixed sea water $\delta^{11}\text{B}$ value for both seasons.

In summary:

This is a high-quality paper that is very well written and easy to understand that will add a lot to our understanding of the interrelation of host zooxanthellae and calcification processes in a changing world.

I ask that the authors will however mention the point of DBL pH (please cite the literature offered here as this defiantly can't be ignored) possibly affecting the Boron fractionation (no need to modify any calculation or graph) thus opening the minds of the readers to the complexity of the process and other possibilities out there, Making the picture wholesome.
all the best

Reviewer #3 (Remarks to the Author)

The manuscript presents a very interesting and novel set of observations of the carbonate chemistry (pH and CO_3) of the calcifying fluid of *Porites* on two reef sites (Davies and Ningaloo) based on geochemical proxies. The data suggest that the pH of the calcifying fluid (cf) goes through a minima during the austral summer and the DIC estimated from Boron-11 pH and the B/Ca elemental ratio of the skeletal material goes through a maxima and attains levels of 2-3 times seawater. The two signals alter the saturation state in opposite directions so that together they tend to smooth out the seasonality. The data are certainly interesting and there is a growing consensus based on a variety of methods that the pH of the CF is elevated by 0.3-0.6 units relative to SW. This is the first study to show that the pH of the CF is dynamic and varies seasonally. There is much less consensus on the DIC or TA of the CF. Cai et al 2016 NCOMMS found that the DIC was slightly less than that of ambient SW based on micro electrode techniques counter to this study which found it to be elevated relative to SW by 2-3 fold. Both methods are subject to uncertainties so it is hard to judge which is more likely to be right. Both studies concluded that their finding of the DIC being high or low relative to SW had important implications for corals response to OA conditions. Neither study measured calcification so their conclusions have to be considered as supposition and not an actual demonstration that calcification rates do

not vary with saturation state while many empirical studies have shown that it does. I think these results should be reported but the conclusions regarding ocean acidification not being important or much less important than bleaching should be softened. Part of the reason for this recommendation is that OA has also been shown to reduce coral recruitment. The impact of OA on all phases of the coral life cycle need to be considered before we conclude that it does not pose a serious threat to coral survival.

Below are our detailed response to the reviewers comments. Changes are shown in highlight together with line numbers. Our responses to reviewers' comments are in blue with the changes in the text highlighted.

Reviewer #1 (Remarks to the Author):

Coral reefs are facing multiple stresses of global warming and ocean acidification (OA). Many works have addressed how corals may respond to OA with a focus on a mechanistic understanding of coral calcification. It is still quite debatable what is the interior pH and DIC in the calcifying fluid (cf) due to the challenge of the work and limitations of various approaches. This work represents an important progress and is novel and timely.

1. This paper uses $\delta^{11}\text{B}$ to derive pH_{cf} and B/Ca geochemical tracer to derive $[\text{CO}_3^{2-}]$ in corals under natural ocean conditions (not cultured/controlled). These authors find that pH_{cf} is about 8.3 to 8.5 (similar to the Allison et al. 2014 paper), which is much lower than the recent report by Cai et al. (2016) based on microelectrodes. Their reported $[\text{CO}_3^{2-}]$ (I guess from omega) are actually quite similar to the microelectrode results. The rest is CO₂SY_S calculation. With a high pH_{cf} the same $[\text{CO}_3^{2-}]_{\text{cf}}$ (and omega) predicts low DIC_{cf} while the opposite is true for low pH_{cf}. Thus these authors find their DIC_{cf} to be 2x-3x of the seawater values. This is not totally new as the Allison et al (2014) paper says similarly. What is novel of this work is that data were obtained from outdoor in the natural reefs and in that anti-phase changes in pH_{cf} and DIC_{cf} were observed. **I feel the authors need to note in the paper clearly the differences and similarities with previous work.** In doing so I believe it is necessary to note that the boron isotope work represents an average of light-dark conditions and even some level of seasonal average while the microelectrode represents a short time scale of light period. Also the microelectrode only measures the cf space under the coral mouth while the geochemical tracers measure average of corals. The authors used the work of Venn et al. (2011) to support their lower pH_{cf}. But as described by Holcomb et al., 2014 they should also recognize the differences between geochemical proxy, microelectrode and pH-dye based measurements. For example, micro-electrode studies made through the centre of coral polyps that were situated at the apex of larger colonies, while Venn et al. worked on the edge of laterally, newly growing microcolonies of coral on a glass slide. There can be significant differences in coral physiology between the top and sides/edges of most corals, including differences in calcification rates, symbiont density, and isotopic values.

With respect to determinations of pH_{cf} to a large extent this therefore reflects differences between the methods. Skeletal proxy measurements reported here are the average of the precipitated aragonite and thus the measured boron isotopes represent an average of pH during light and dark calcification; however, it is an average that is weighted by the diurnally dependent rates of calcification. Given that rates of daytime calcification are ~3 times higher than night-time calcification on average (Gattuso et al. 1999), then ~75% of the measured $\delta^{11}\text{B}$ value is representing conditions within the calcifying fluid under daytime conditions. Therefore we expect the values of $\delta^{11}\text{B}$ we report should be mostly reflect the pH_{cf} observed under well-lit, albeit somewhat lower

lower than in-situ methods. We note that the night-time decline in pH_{cf} observed by Al-Horani (2003) is likely the result of unnatural levels of carbon limitation as a result of that author working on coral fragments under poorly lit laboratory conditions (at least by in-situ standards). Therefore, there is good reason to believe that our time- and calcification-averaged measurements of $\delta^{11}\text{B}$ -derived pH_{cf} are representative of the general conditions under which calcification occurs.

These differences are already thoroughly described previously (Holcomb, Venn et al. 2014) $\delta^{11}\text{B}$ versus in-situ con-focal pH_{cf} (Al-Horani, Al-Moghrabi et al. 2003, Venn, Tambutte et al. 2013) and hence not elaborated further in this paper.

In response to the request to compare similarities and differences with previous work (Allison et al. 2014 paper and Cai et al., 2016) we have made the following changes/additions in the text.

Lines 76-79. For determination of the carbonate ion concentrations $[\text{CO}_3^{2-}]_{\text{cf}}$ in the calcifying fluid we use the combined $\delta^{11}\text{B}$ -B/Ca proxy (Holcomb, DeCarlo et al. 2016). The application of the $\delta^{11}\text{B}$ -B/Ca carbonate ion proxy has now been made possible by recent experimental measurements of the B/Ca carbonate ion distribution coefficient (Holcomb, DeCarlo et al. 2016), a major limitation of previous studies (Allison, Cohen et al. 2014) (see Methods).

Lines 142-151. Whilst the absolute levels of enhanced Ω_{cf} are not dissimilar to previous qualitative estimates (Holcomb, Cohen et al. 2009, McCulloch, Trotter et al. 2012), the finding of significantly higher (x2.0 to x3.2 seawater) DIC_{cf} is not generally consistent with recent micro-sensor (Cai, Ma et al. 2016) measurements. This difference may reflect the intrinsic limitations (Allemand, Tambutté et al. 2011) of using probes that are 15-20 μm wide to measure the chemistry of the narrow and irregular sub-micrometric (1-10 μm) calcifying region. Additionally different probes are required for separate measurements of pH_{cf} and $[\text{CO}_3^{2-}]_{\text{cf}}$, introducing further uncertainty, especially given that the large spatial variability of in-situ measured CO_3^{2-} and hence inferred DIC_{cf} (~x1.4 to x4.2 seawater). Finally, and most importantly, regardless of the method employed, we find that measurements conducted under controlled, static laboratory conditions, are unlikely to be representative of *natural* reef conditions due to the interactive, anti-phase dynamics of pH_{cf} and DIC_{cf} up-regulation described here.

Lines 324-325 Where $K_D = 0.00297 \exp(-0.0202 [\text{H}^+]_{\text{T}})$ and for typical calcifying fluid pH_{cf} values $K_D \sim 0.0027$, an order of magnitude higher than a previous estimate (Allison, Cohen et al. 2014).

2. I feel one critical question is whether the $[\text{CO}_3^{2-}]_{\text{cf}}$ geochemical tracer method is totally independent from the pH_{cf} tracer method. In p.2 (Results), the paper says "The $\delta^{11}\text{B}$ acts as a proxy for pH_{cf} while the newly developed combination of B/Ca and $\delta^{11}\text{B}$ now provides a proxy for the carbonate ion concentration $[\text{CO}_3^{2-}]_{\text{cf}}$ of the calcifying fluid" According to this, it seems to me that these approaches are not independent; they both rely on $\delta^{11}\text{B}$ data. If so, then, the assumptions in deriving pH_{cf} are carried over into the $[\text{CO}_3^{2-}]_{\text{cf}}$. I checked the equations in p.10 and the Chemical Geology paper and do not see that pH_{cf} is needed in calculating $[\text{CO}_3^{2-}]_{\text{cf}}$. Is this because both K_D and $[\text{B}(\text{OH})_4]_{\text{cf}}$ contains/requires pH_{cf}? Then, please comment if pH_{cf} is under-estimated, which direction will it affect the estimation of $[\text{CO}_3^{2-}]_{\text{cf}}$ and DIC_{cf} ? Some explanation/clarification is

needed here. I remember in the Allison et al. 2014 paper, when they used the B/Ca tracer to derive DIC, they had to make an assumption whether B is precipitated into the CaCO₃ with HCO₃⁻ or with CO₃²⁻. When the assumption is the precipitation with HCO₃⁻, one would get high DIC, but if it is with CO₃²⁻, one would get low DIC. I suspect this is a related issue if indeed the [CO₃²⁻] is derived as a combination of B/Ca and δ¹¹B.

The reviewer is correct in that we have calculated the concentration of carbonate within the calcifying fluid (cf) directly from the B/Ca of the skeletal material and concentration of borate in the CF, the latter being dependent on pH_{cf} and can vary by ±20% between a pH_{cf} of 8.3 to 8.6 (the range in which most tropical coral fall). Although the K_D for B/Ca is also theoretically dependent on pH, it is actually quite insensitive over this same observed range in pH_{cf} varying by less than ±3%. Therefore we expect calculations of [CO₃²⁻]_{cf} not to be very sensitive to the pH dependency of K_D_B/Ca. This has now been explicitly mentioned

The discrepancy between our results and those predicted by Allison et al. (2014) and those reported by Holcomb et al. (2016) are the result of the much more reliable determination of K_D_B/Ca from the inorganic precipitation experiments of Holcomb et al. (2016) from which direct measurements B/Ca, [carb], and [borate] which made. The Holcomb et al. (2016) determinations are an order of magnitude higher than value estimated by Allison et al. (2014). Allison's estimate was based on the B/Ca of an 'inorganic precipitate' on fossil (13 ka) coral together with estimates the pore water alkalinity (and B/Ca) from which the precipitate supposedly formed. **This is clearly a highly dubious estimate and it is not surprising to be in error by an order of magnitude.** Using the properly determined K_D from Holcomb's far more rigorous experimentally controlled approach compared to Allison's equivalent scenario 1 her DIC_{cf} values are x2-3.5 times lower and hence less seawater. In an attempt to overcome this inconsistency it now appears that Allison evoked an a-hoc bicarbonate substitution model (described as scenario 2) which is inconsistent with the K_D concept and has no proper chemical basis. In summary it is now clear that the results of Allison et al. (2014) were affected by an unreliable indirect estimate of K_D (B/Ca) and that there is no need to invoke the involvement of bicarbonate which, in itself has no chemical justification. If bicarbonate were a key substrate in the formation of carbonate minerals than the concept of saturation state as it is currently defined (on carbonate activity alone) would have little or no meaning to the formation of carbonate minerals.

Since our work is based on proper measurements of K_D and is consistent with borate/carbonate ion substitution we have made the following very tempered changes/additions. We have thus minimised explicit criticism of the Allison et al., (2014) study, which is in any case now redundant (and as mentioned above clearly fraught with serious errors etc which are beyond the scope of this paper).

Lines 76-79 (see previous). **For determination of the carbonate ion concentrations [CO₃²⁻]_{cf} in the calcifying fluid we use the combined δ¹¹B-B/Ca proxy(Holcomb, DeCarlo et al. 2016) whose application has now been made possible by recent(Holcomb, DeCarlo et al. 2016) experimental measurements of the B/Ca carbonate ion distribution coefficient, a major limitation of previous studies(Allison, Cohen et al. 2014) (see Methods).**

Lines 324-325 Where $K_D = 0.00297 \exp(-0.0202 [H^+]_T)$ and for typical calcifying fluid pH_{cf} values $K_D \sim 0.0027$, an order of magnitude higher than a previous estimate (Allison, Cohen et al. 2014).

Lines 330-333 We also note that our use of a reliable experimentally determined K_D is now consistent with substitution of borate with carbonate ion, rather than the previously inferred (Allison, Cohen et al. 2014) substitution with bicarbonate ion, the latter assumption effectively negating the role of carbonate saturation state on calcification.

3. At the relatively low pH_{cf} and the 2-3X DIC, what is the $[CO_2]$ concentration inside the cf? Is it still realistic that enough respiratory CO_2 can diffuse from the coelenteron into the cf? The very high internal DIC (low pH_{cf}) indicates relatively high $[CO_2]$ inside the cf. Thus, does CO_2 molecular still have the needed gradient to diffuse in to supply the calcification rate? The authors should check if the high DIC result is internally consistent.

The reviewer makes an interesting point here. By our calculations, the concentration of CO_2 in the CF would be around 8 and 17 at those same temperatures, respectively; whereas the concentration of CO_2 in the coelenteron would be 22 and 19 at these temperatures assuming its pH were ~ 7.8 . However, these calculations don't provide the full story since 1) this CO_2 more likely comes from mitochondria whose local $[CO_2]$ are likely far higher (and as yet unknown) than found in the coelenteron rather than the coelenteron itself and 2) it assumes that the supply of carbon of metabolically generated carbon to the CF is primarily in the form of CO_2 . There is recent evidence that corals can actively transport bicarbonate into the CF (Zoccola et al. 2015) although the potential for bicarbonate transport to provide a mechanism of DIC supply to the CF is already well-known to the literature (Allemand et al. 2011). We believe that the active transport of bicarbonate is more likely to provide the necessary mechanism by which coral can support the relatively high rates of metabolically driven calcification than passive diffusion of CO_2 across the calcoblastic cells; however, it is not our intent to resolve this issue in this paper. This is why we are not stating that one particular substrate (CO_2 or bicarbonate) is more important than the other. Thus, any calculation of $[CO_2]$ within the coelenteron will not provide any reliable constraints about what our current calculations of DIC_{cf} imply about the feasibility of carbon transport into the CF. That issue is plagued from too many other more important uncertainties.

Reading notes (or minor points)

p.1, while nothing really wrong, it's first time I heard anyone say "dissolved inorganic carbonate (DIC)". We normally say dissolved inorganic carbon (DIC) though the meaning is the same that is the total concentration of all dissolved inorganic carbon species. I don't think it is necessary the authors create new name.

p.1 and several other places: the citation # after CO_2 should be CO_2 ref14 (not CO_2 14) to avoid confusion.

p.1, for Ca ions, a "2+" should be attached to it (Ca^{2+}).

p.1. fix this “It has also been long been recognized”

Figure 1 caption, delete repeat: Transfer of DIC into the into the

Figure 1. although the fig caption says “Transfer of DIC into the into the sub-calicoblastic space may occur via diffusion of CO₂ and/or by HCO₃⁻ - pumping via bicarbonate anion transporters” the graphic illustration (right middle part) only shows HCO₃⁻ transport. The CO₂ diffusive transport is not clear. This should be the main transport DIC into the coral interior space to support calcification as argued by Allison et al. 2014 and Cai et al. 2016.

A CO₂ route is shown in Figure 1 although not highlighted. Furthermore we don't believe that Cai et al. 2016 have presented enough evidence to support their argument that CO₂ is the main form of DIC being transported into the CF since neither study actually measured synoptic rates of carbon transport or calcification. Cai et al. (2016) calculated the total flux of CO₂ by passive diffusion only which is only one of several pathways by which carbon supply to the cf has been proposed. Cai et al. (2016) further based their argument on the importance of passive CO₂ diffusion on the assertion that the energetic cost of pumping protons is high although the logic is unclear since the reliance of calcification on CO₂ supply would require far more proton pumping than the reliance on bicarbonate transport. Either way, McCulloch et al. (2012) specifically de-bunked the idea that pH up-regulation was thermodynamically costly given the high rates of photosynthesis that have been measured in coral over the past several decades. **Finally we emphasise that this is not the thrust of the paper and therefore reframed from being overly specific about the DIC transport mechanisms.**

p.2, Results

the paper says “The d11B acts as a proxy for pH_{cf} 5,7 while the newly developed⁷ combination of B/Ca and d11B now provides a proxy for the carbonate ion concentration [CO₃²⁻]_{cf} of the calcifying fluid” My question is—do you do both B/Ca and d11B to derive [CO₃²⁻]_{cf}? Or just B/Ca? If you need both, then, I would argue the two methods are not strictly independent, that is all the assumptions in deriving pH_{cf} are carried over into the [CO₃²⁻]_{cf}. I checked the equation in p.10 (it should also be numbered) and the Chemical Geology paper and do not see pH_{cf} as a needed information in calculating [CO₃²⁻]_{cf}.

See earlier comment

p.4, top lines: “The explanation for this unexpectedly large range in seasonal pH_{cf} found under ‘real-world’ conditions becomes especially apparent” this sentence needs modification as I do not see any “explanation” let alone it be “apparent”. If what follows “The DIC_{cf} thus also shows” provides the explanation, then, some transition words should be used and “thus” should be deleted.

Change made Lines 101-131 **The explanation for this unexpectedly large range in seasonal pH_{cf} present under natural reef conditions becomes apparent from the exceptionally strong, inverse strong, inverse correlations between pH_{cf} and DIC_{cf} ($r^2 = 0.88$ to 0.94) present at the colony level (Figure 3a,b). Here DIC_{cf} reaches its highest values in summer ($\times 2.0$ to $\times 3.2$ higher than ambient seawater) and lowest values in winter, whereas pH_{cf} shows the opposite pattern.**

p.4. you said “Whilst the absolute levels of enhanced Ω_{cf} are not dissimilar to previous estimates^{4,28}, the finding of significantly higher ($\sim x2$ to $x3$ seawater) DIC_{cf} is not consistent with recently reported micro-sensor CO₃₂₋ measurements²² pointing to limitations of such intrusive measurements into the tight sub-micrometric (1-10 μm) calcifying region².” I think some clarification is needed here. First, as far as [CO₃₂₋] (or Ω_{cf}) is concerned, the range reported here (4 to 6 times of seawater) is not different from the direct microelectrode measured [CO₃₂₋] reported in ref 22 by Cai et al. 2016. What is really different is pH_{cf}. This fact should be pointed out. Because of a much higher pH_{cf} in Cai et al., their calculated DIC_{cf} is very low (1X seawater). Second, what Cai et al. reported in lab was under light condition and nearly instantaneous. This can be very different from the seasonal averaged pH_{cf} reported here. I agree there are limitations of the microelectrode approach, but simply say “pointing to limitations of such intrusive measurements into the tight sub-micrometric (1-10 μm) calcifying region²” is probably not enough. Possible real differences should be mentioned.

Please see our earlier comment about the time- and calcification-averaging of pH_{cf} by d11B measurements.

p.5, bottom, “Together these observations also argue against the possibility that the reduction of pH_{cf} in summer is due to the passive feed-back from higher rates of calcification producing more protons thereby lowering pH_{cf}. Whilst this possibility cannot yet be entirely excluded the higher production rates of zooxanthellae derived metabolites in the summer to drive Ca-ATPase activity also suggest that the season up-regulation of elevated pH_{cf} is due direct to growth rate control rather than from limitations in the Ca-ATPase H⁺ pumping.”

I just don't see how you can distinguish one from the other. But you can use the ratio of changes in DIC and TALK to say something.

We agree with the reviewer that we cannot prove that the decline in pH_{cf} in summer is the result of active regulation of Ω_{cf} by the coral rather than the passive result of rates of cation exchange simply not being able to maintain pH_{cf} as high as in winter due to much higher rates of alkalinity removal within the CF due to calcification. Our original wording attempted to reflect this ambiguity, but perhaps not enough to the reviewer's liking. We have re-written the sentence accordingly.

Lines 200-204 Thus while this possibility cannot yet be entirely excluded, the higher production rates of zooxanthellae derived metabolites that are presumeably available in the summer to facilitate enhanced Ca-ATPase activity, also suggest that the lower summer levels of pH_{cf} is not due to intrinsic limitations in the Ca-ATPase H⁺ pumping, but rather physiologically-based growth rate control.

p.x, 6 lines above Methods: “Although this process is not directly influenced by ocean acidification, it is however highly vulnerable to thermal stress.” To say a process is “vulnerable” doesn't really sound right. modify.

See lines 229-231. Although in mature coral colonies the maintenance of elevated but near constant Ω_{cf} is not directly influenced by ocean acidification, it is however highly susceptible to thermal stress.

Refs:

Allison, N.; Cohen, I.; Finch, A. A.; Erez, J.; Tudhope, A. W., Corals concentrate dissolved inorganic carbon to facilitate calcification. Nat Commun 2014, 5.

Cai, W.-J.; Ma, Y.; Hopkinson, B. M.; Grottoli, A. G.; Warner, M. E.; Ding, Q.; Hu, X.; Yuan, X.; Schoepf, V.; Xu, H.; Han, C.; Melman, T. F.; Hoadley, K. D.; Pettay, D. T.; Matsui, Y.; Baumann, J. H.; Levas, S.; Ying, Y.; Wang, Y., Microelectrode characterization of coral daytime interior pH and carbonate chemistry. Nat Commun 2016, 7.

Venn, A.; Tambutté, E.; Holcomb, M.; Allemand, D.; Tambutté, S., Live tissue imaging shows reef corals elevate pH under their calcifying tissue relative to seawater. PLoS ONE 2011, 6, (5), e20013.

Reviewer #2 (Remarks to the Author):

This article focuses on the active control of the coral host and photo symbionts on the chemical properties (DIC and pH) of the calcifying fluid (cf) under natural conditions. Using state of the art methods: Boron isotopic pH proxy and B/Ca constraints on calcifying fluid DIC concentrations, the authors reconstructed the latter properties in correlation to the natural conditions at sea.

Main find: antithetic relationships between DIC (dissolved inorganic carbon) and pH of the corals calcifying fluid (cf). while in the summer, under higher temperatures DIC_{cf} was 3.2 seawater the pH was 8.3, in the winter DIC_{cf} was 2.0 seawater the pH was 8.5. In both cases maintaining W_{cf} ~x5 seawater.

The conceptual main conclusion and in my eyes the most important is: "These findings are in marked contrast to artificial experiments and show that up-regulation of pH_{cf} occurs largely independent of changes in seawater carbonate chemistry and hence ocean acidification." General comments to authors: this work is of the upmost high end both technically and conceptually to try and elucidate the different biochemical and chemical pathways that influence the calcification process in corals. It is very well presented and the writing allows a wide range of readers to take part in this fascinating discussion. It was indeed a pleasure to read and I am happy if my comments will augment this article in its final form.

Comments to the authors

First paragraph:

Not sure if the authors meant to put in the word "but" in the sentence perhaps "and" is also ok: "These opposing changes in DIC_{cf} and pH_{cf} are shown to maintain highly oversaturated (W_{cf} ~x5 seawater) but relatively stable levels of carbonate saturation, the key parameter controlling the rate of coral calcification⁴"

In the following sentence where the authors make the claim “These findings are in marked contrast to artificial experiments a...” I think that the authors should list the references to these articles.

We agree with the spirit of the reviewers’ suggestion, but so many experiments have been conducted on the effects of OA on coral growth that the number of relative citations could fill an entire line of the Abstract. Instead, we have cited the meta-analysis of Chan and Connolly (2012).

Second section / intro and the rest. My main issue with this section is rooted in the described (fig 1) mechanism for DIC transport and pH homeostasis. The authors chose to calculate the pH in the calcifying fluid (pH_{cf}) using the $\delta^{11}\text{B}$ value of sea water - $\delta^{11}\text{B}$ value of the coral skeleton. This calculation excludes the effect of the diffusion boundary layer (DBL) on the isotopic composition of the actual pool the coral incorporates into its skeleton. In 1995 Michael Kuhl et al. published a paper (Mar. Ecol. Prog. Ser, Vol. 117: 159-172.1995.) where for the first time he introduced a pH profile of the DBL. This publication followed an earlier one by Shashar et al. 1993 Biol. Bull. 185: 455-46 1. (1993), who presented the oxygen gradient in a coral DBL. In 2005 El Horani published another work that followed the DBL under changing temperatures establishing how in close proximity to the corals surface the changes in pH are pronounced (a Δ of 0.9 pH units). Lastly a paper from the Tchernov (Biogeosciences, 12,5677–5687, 2015) also showed the effect of symbiont photosynthesis rates on the DBL pH and finally on the $\delta^{11}\text{B}$ value of the coral skeleton. The pH near the coral surface will influence the pool of Boron that will finally be incorporated via the cf into the skeleton. The pH in the cf will further influence the Boron isotopic composition via discrimination based on the local pH. Therefore we are facing a two-step discrimination process (as I understand).

Although the work by Shashar et al. (1993) and Kuhl et al. (1995) were seminal in introducing the existence of chemical boundary layers to the reef community so many years ago, we believe that all of the boron being transferred to the CF is occurring as part of the normal paracellular transport of bulk seawater that proceeds the initiation of calcification and whose abundance is therefore dependent only on the salinity of the seawater. This form of bulk transport is not subject to the same kind of isotopic fractionation that passive diffusion would cause as it should preserve the isotopic composition of the bulk boron, regardless of the pH of the seawater when it is transported into the CF. After being transported, this fixed isotopic composition (39.61 ‰) is then partitioned between the borate and boric acid pools within the CF according to the pH_{cf}. The reason passive diffusion of boron is not further required during calcification is because its partition coefficient (KD) relative to calcium is so low ($\sim 3 \times 10^{-3}$). Thus, little boron is actually removed during calcification meaning that there is no significant reduction in the total concentration of boron from ambient seawater actually occurs. Furthermore, there is no as yet identified pathway for the active or passive diffusion of boron into the CF so we don’t think transport-driven fractionation of the pH_{cf} is an important process. We doubt that borate and boric acid are diffusing into the CF according to their concentration at the surface that, if true, would imply that fractionation is a two-step process. Furthermore, the highly linear arrays of pH_{cf} vs. pH_{sw} across a wide range of tropical and temperate coral as demonstrated by Trotter et al. (2011) further suggest that the relationship between pH_{cf} and $\delta^{11}\text{B}$ is not being affected by spatially and temporally variable boundary layer dynamics and more by the straight forward process of internal pH up-regulation. If boundary layer dynamics were that important, then those pH_{cf}-pH_{sw} relationships should have been far noisier they were

observed to be. Instead, they are very highly correlated ($r^2 > 0.91$ for all species with many $r^2 > 0.99$).

In the methods section equation 2 is presented: (cant paste it here) but I am sure you have it :)

$\delta^{11}\text{B}_{\text{sw}} = 39.61 \text{ ‰}$ is referred from (Foster, G. L., Pogge von Strandmann, P. A. E. & Rae, J. W. B.) as a fixed value thus not considering the boundary layer (depending on the non-constant photosynthesis \ respiration rates as described by relevant literature). From this equation alone it is evident that marked changes in seawater Boron isotopes values will affect the pH_{cf}.

We believe the uptake of boron through the transport of bulk seawater preserves the bulk isotopic composition of the seawater regardless of pH. Finally we note that there is excellent agreement between pH_{cf} measurements using $\delta^{11}\text{B}$ and confocal microscopy (see Holcomb et al., 2014) and see earlier comment.

With all that said, I still think that the work presented merits publication in this prestigious journal as it shows the mechanistically coupled alga -host biochemical and chemical processes influencing the calcification pathway in corals. The summer and winter differences are quite convincing and are indeed a valuable contribution to our field. From a plethora of manuscripts we gather that during summer the photosynthetic rate is enhanced in agreement with the authors statements (“was found during summer growth, consistent with thermal/light enhancement of metabolically (zooxanthellae) derived DIC, while the highest pH_{cf} (~8.5) occurred in winter during periods of low DIC_{cf} (~2 seawater).”) that does imply that the coral does indeed actively controls the cf’s pH otherwise it would be in direct correlation with temperature (high temperature high Photosynthetic rate leading to high cf pH) however the authors show a reverse situation. We still have to consider that this was calculating with a fixed sea water d¹¹B value for both seasons.

In summary:

This is a high-quality paper that is very well written and easy to understand that will add a lot to our understanding of the interrelation of host zooxanthellae and calcification processes in a changing world.

I ask that the authors will however mention the point of DBL pH (please cite the literature offered here as this defiantly can’t be ignored) possibly affecting the Boron fractionation (no need to modify any calculation or graph) thus opening the minds of the readers to the complexity of the process and other possibilities out there, Making the picture wholesome. all the best

We appreciate the reviewer’s firm belief in the importance of diffusional boundary layers to the exchange of metabolites in benthic reef organisms given that one of our authors has dedicated considerable the physically driven mechanics of convective mass transfer under the kind of realistic wave- and current-driven flows over rough, natural topographies that are difficult to reproduce in scaled-down, bench-top experiments (see Falter, J. L., R. J. Lowe, and Z. Zhang. 2016. *Towards a universal mass-momentum transfer relationship for predicting nutrient uptake and metabolite exchange in benthic reef communities*. Geophys. Res. Lett. 1–9). We just don’t believe that passive

diffusion is as important to the uptake and transport of boron to the CF as it is for say oxygen, CO₂, and dissolved nutrients. We think the reviewer makes a much more compelling case for the interpretation of skeletal $\delta^{13}\text{C}$ than $\delta^{11}\text{B}$.

Reviewer #3 (Remarks to the Author):

The manuscript presents a very interesting and novel set of observations of the carbonate chemistry (pH and CO₃) of the calcifying fluid of Porites on two reef sites (Davies and Ningaloo) based on geochemical proxies. The data suggest that the pH of the calcifying fluid (cf) goes through a minima during the austral summer and the DIC estimated from Boron-11 pH and the B/Ca elemental ratio of the skeletal material goes through a maxima and attains levels of 2-3 times seawater. The two signals alter the saturation state in opposite directions so that together they tend to smooth out the seasonality. The data are certainly interesting and there is a growing consensus based on a variety of methods that the pH of the CF is elevated by 0.3-0.6 units relative to SW. This is the first study to show that the pH of the CF is dynamic and varies seasonally. There is much less consensus on the DIC or TA of the CF. Cai et al 2016 NCOMMS found that the DIC was slightly less than that of ambient SW based on micro electrode techniques counter to this study which found it to be elevated relative to SW by 2-3 fold. Both methods are subject to uncertainties so it is hard to judge which is more likely to be right. Both studies concluded that their finding of the DIC being high or low relative to SW had important implications for corals response to OA conditions. Neither study measured calcification so their conclusions have to be considered as supposition and not an actual demonstration that calcification rates do not vary with saturation state while many empirical studies have shown that it does. I think these results should be reported but the conclusions regarding ocean acidification not being important or much less important than bleaching should be softened. Part of the reason for this recommendation is that OA has also been shown to reduce coral recruitment. The impact of OA on all phases of the coral life cycle need to be considered before we conclude that it does not pose a serious threat to coral survival.

We agree that there other ways that OA can affect the ecology of reef-building coral. We've added the following sentences (lines 229-231) Thus, although rising levels of pCO₂ can have adverse effects on the recruitment and growth of juvenile corals (Albright and Langdon 2011, de Putron, McCorkle et al. 2011, Tambutté, Venn et al. 2015, Foster, Falter et al. 2016), especially those lacking robust internal carbonate chemistry regulatory mechanisms, extreme thermal stress is detrimental to all symbiont bearing corals (Randall and Szmant 2009, Chua, Leggat et al. 2013) regardless of their life-history phase.

With additional references (35-38).

Yours sincerely
Malcolm McCulloch

References for comments

- Al-Horani, F. A., S. M. Al-Moghrabi and D. de Beer (2003). "Microsensor study of photosynthesis and calcification in the scleractinian coral, *Galaxea fascicularis*: active internal carbon cycle." Journal of Experimental Marine Biology and Ecology **288**((1)): 1-15.
- Albright, R. and C. Langdon (2011). "Ocean acidification impacts multiple early life history processes of the Caribbean coral *Porites astreoides*." Global Change Biology **17**(7): 2478-2487.
- Allemand, D., E. Tambutté, D. Zoccola and S. Tambutté (2011). Coral calcification, cells to reefs Coral reefs: an ecosystem in transition. Z. Dubinsky and N. Stambler, Springer, Berlin, Germany. **III**: 119-150.
- Allison, N., I. Cohen, A. A. Finch, J. Erez and A. W. Tudhope (2014). "Corals concentrate dissolved inorganic carbon to facilitate calcification." Nature communications **5**.
- Cai, W.-J., Y. Ma, B. M. Hopkinson, A. G. Grottoli, M. E. Warner, Q. Ding, X. Hu, X. Yuan, V. Schoepf, H. Xu, C. Han, T. F. Melman, K. D. Hoadley, D. T. Pettay, Y. Matsui, J. H. Baumann, S. Levas, Y. Ying and Y. Wang (2016). "Microelectrode characterization of coral daytime interior pH and carbonate chemistry." Nature Communications **7**: 11144.
- Chua, C. M., W. Leggat, A. Moya and A. H. Baird (2013). "Temperature affects the early life history stages of corals more than near future ocean acidification." Marine Ecology Progress Series **475**: 85-92.
- de Putron, S. J., D. C. McCorkle, A. L. Cohen and A. Dillon (2011). "The impact of seawater saturation state and bicarbonate ion concentration on calcification by new recruits of two Atlantic corals." Coral Reefs **30**(2): 321-328.
- Foster, T., J. L. Falter, M. T. McCulloch and P. L. Clode (2016). "Ocean acidification causes structural deformities in juvenile coral skeletons." Science advances **2**(2): e1501130.
- Holcomb, M., A. L. Cohen, R. I. Gabbitov and J. L. Hutter (2009). "Compositional and morphological features of aragonite precipitated experimentally from seawater and biogenically by corals." Geochimica et Cosmochimica Acta **73**(14): 4166-4179.
- Holcomb, M., T. M. DeCarlo, G. A. Gaetani and M. McCulloch (2016). "Factors affecting B/Ca ratios in synthetic aragonite." Chemical Geology **437**: 67-76.
- Holcomb, M., A. A. Venn, E. Tambutte, S. Tambutte, D. Allemand, J. Trotter and M. McCulloch (2014). "Coral calcifying fluid pH dictates response to ocean acidification." Sci. Rep. **4**: 5207-5211.
- McCulloch, M. T., J. A. Trotter, J. Falter and P. Montagna (2012). "Coral resilience to ocean acidification and global warming through pH up-regulation " Nature Climate Change **2**: 623-627.
- Randall, C. J. and A. M. Szmant (2009). "Elevated temperature affects development, survivorship, and settlement of the elkhorn coral, *Acropora palmata* (Lamarck 1816)." The Biological Bulletin **217**(3): 269-282.
- Tambutté, E., A. Venn, M. Holcomb, N. Segonds, N. Techer, D. Zoccola, D. Allemand and S. Tambutté (2015). "Morphological plasticity of the coral skeleton under CO₂-driven seawater acidification." Nature communications **6**.
- Venn, A. A., E. Tambutte, M. Holcomb, J. Laurent, D. Allemand and S. Tambutte (2013). "Impact of seawater acidification on pH at the tissue-skeleton interface and calcification in reef corals." Proc Natl Acad Sci U S A **110**(5): 1634-1639.

Reviewers' Comments:

Reviewer #1 (Remarks to the Author)

I appreciate that the authors seriously addressed my concerns. While I do not fully agree with the authors, in particular regarding the relative importance of CO₂ vs HCO₃⁻ transport, I feel the arguments and limited modifications made by the authors are reasonable. I particularly like the way the authors addressed the difference between their work and that of the Allison et al.—point out the differences in a kind tone.

Reviewer #2 (Remarks to the Author)

Although the work by Shashar et al. (1993) and Kulh et al. (1995) were seminal in introducing the existence of chemical boundary layers to the reef community so many years ago, we believe that all of the boron being transferred to the CF is occurring as part of the normal paracellular transport of bulk seawater that precedes the initiation of calcification and whose abundance is therefore dependent only on the salinity of the seawater. This form of bulk transport is not subject to the same kind of isotopic fractionation that passive diffusion would cause as it should preserve the isotopic composition of the bulk boron, regardless of the pH of the seawater when it is transported into the CF. After being transported, this fixed isotopic composition (39.61 ‰) is then partitioned between the borate and boric acid pools within the CF according to the pH_{cf}. The reason passive diffusion of boron is not further required during calcification is because its partition coefficient (KD) relative to calcium is so low (~3x10⁻³). Thus, little boron is actually removed during calcification meaning that there is no significant reduction in the total concentration of boron from ambient seawater actually occurs. Furthermore, there is no as yet identified pathway for the active or passive diffusion of boron into the CF so we don't think transport-driven fractionation of the pH_{cf} is an important process. We doubt that borate and boric acid are diffusing into the CF according to their concentration at the surface that, if true, would imply that fractionation is a two-step process. Furthermore, the highly linear arrays of pH_{cf} vs. pH_{sw} across a wide range of tropical and temperate coral as demonstrated by Trotter et al. (2011) further suggest that the relationship between pH_{cf} and δ₁₁B is not being affected by spatially and temporally variable boundary layer dynamics and more by the straight forward process of internal pH up-regulation. If boundary layer dynamics were that important, then those pH_{cf}-pH_{sw} relationships should have been far noisier they were observed to be. Instead, they are very highly correlated (r² > 0.91 for all species with many r² > 0.99). I appreciate the author's detailed response: passive diffusion is not the issue at all. Bulk uptake will be similarly affected by the pH at the boundary layer since the **source of the boron is always** from the immediate vicinity of the corals outer membrane that is governed by a pH régime that is dependent on photosynthesis.

“this fixed isotopic composition (39.61 ‰)” thus this is not a fixed composition and in this case may diverge depending on the DBL light dependent pH.

However I am not going to prevent the publication on this account, as I wrote, it's an excellent paper.